# What Uncertainties Do We Need in Bayesian Deep Learning for Computer Vision?

**Alex Kendall**
University of Cambridge
agk34@cam.ac.uk

**Yarin Gal**
University of Cambridge
yg279@cam.ac.uk

## Abstract

There are two major types of uncertainty one can model. *Aleatoric* uncertainty captures noise inherent in the observations. On the other hand, *epistemic* uncertainty accounts for uncertainty in the model – uncertainty which can be explained away given enough data. Traditionally it has been difficult to model epistemic uncertainty in computer vision, but with new Bayesian deep learning tools this is now possible. We study the benefits of modeling epistemic vs. aleatoric uncertainty in Bayesian deep learning models for vision tasks. For this we present a Bayesian deep learning framework combining input-dependent aleatoric uncertainty together with epistemic uncertainty. We study models under the framework with per-pixel semantic segmentation and depth regression tasks. Further, our explicit uncertainty formulation leads to new loss functions for these tasks, which can be interpreted as learned attenuation. This makes the loss more robust to noisy data, also giving new state-of-the-art results on segmentation and depth regression benchmarks.

## 1  Introduction

Understanding what a model does not know is a critical part of many machine learning systems. Today, deep learning algorithms are able to learn powerful representations which can map high dimensional data to an array of outputs. However these mappings are often taken blindly and assumed to be accurate, which is not always the case. In two recent examples this has had disastrous consequences. In May 2016 there was the first fatality from an assisted driving system, caused by the perception system confusing the white side of a trailer for bright sky [1]. In a second recent example, an image classification system erroneously identified two African Americans as gorillas [2], raising concerns of racial discrimination. If both these algorithms were able to assign a high level of uncertainty to their erroneous predictions, then the system may have been able to make better decisions and likely avoid disaster.

Quantifying uncertainty in computer vision applications can be largely divided into regression settings such as depth regression, and classification settings such as semantic segmentation. Existing approaches to model uncertainty in such settings in computer vision include particle filtering and conditional random fields [3, 4]. However many modern applications mandate the use of *deep learning* to achieve state-of-the-art performance [5], with most deep learning models not able to represent uncertainty. Deep learning does not allow for uncertainty representation in regression settings for example, and deep learning classification models often give normalised score vectors, which do not necessarily capture model uncertainty. For both settings uncertainty can be captured with *Bayesian deep learning* approaches – which offer a practical framework for understanding uncertainty with deep learning models [6].

In Bayesian modeling, there are two main types of uncertainty one can model [7]. *Aleatoric* uncertainty captures noise inherent in the observations. This could be for example sensor noise or motion noise, resulting in uncertainty which cannot be reduced even if more data were to be collected. On the other hand, *epistemic* uncertainty accounts for uncertainty in the model parameters – uncertainty

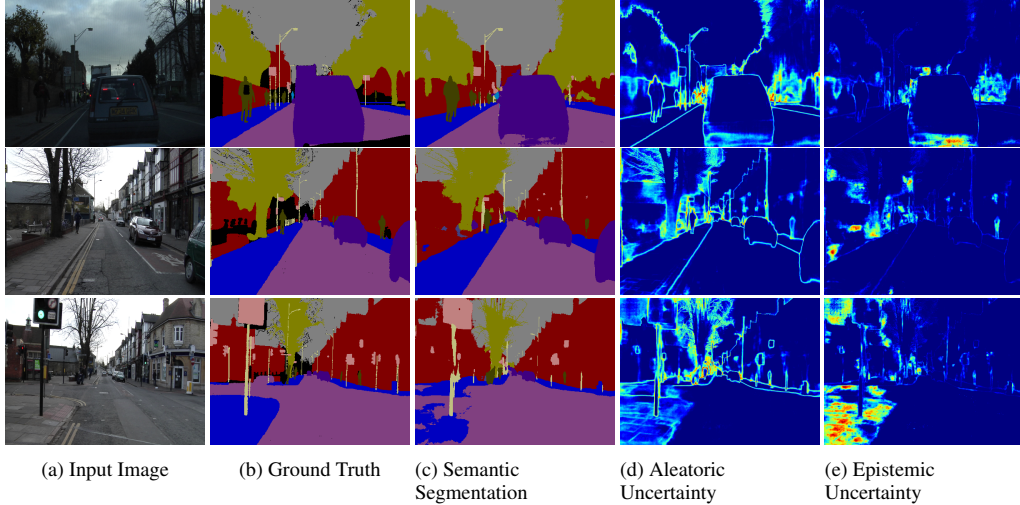

| (a) Input Image | (b) Ground Truth | (c) Semantic Segmentation | (d) Aleatoric Uncertainty | (e) Epistemic Uncertainty |

Figure 1: **Illustrating the difference between aleatoric and epistemic uncertainty** for semantic segmentation on the CamVid dataset [8]. *Aleatoric* uncertainty captures noise inherent in the observations. In (d) our model exhibits increased aleatoric uncertainty on object boundaries and for objects far from the camera. *Epistemic* uncertainty accounts for our ignorance about which model generated our collected data. This is a notably different measure of uncertainty and in (e) our model exhibits increased epistemic uncertainty for semantically and visually challenging pixels. The bottom row shows a failure case of the segmentation model when the model fails to segment the footpath due to increased epistemic uncertainty, but not aleatoric uncertainty.

which captures our ignorance about which model generated our collected data. This uncertainty can be explained away given enough data, and is often referred to as *model uncertainty*. Aleatoric uncertainty can further be categorized into *homoscedastic* uncertainty, uncertainty which stays constant for different inputs, and *heteroscedastic* uncertainty. Heteroscedastic uncertainty depends on the inputs to the model, with some inputs potentially having more noisy outputs than others. Heteroscedastic uncertainty is especially important for computer vision applications. For example, for depth regression, highly textured input images with strong vanishing lines are expected to result in confident predictions, whereas an input image of a featureless wall is expected to have very high uncertainty.

In this paper we make the observation that in many big data regimes (such as the ones common to deep learning with image data), it is most effective to model aleatoric uncertainty, uncertainty which cannot be explained away. This is in comparison to epistemic uncertainty which is mostly explained away with the large amounts of data often available in machine vision. We further show that modeling aleatoric uncertainty alone comes at a cost. Out-of-data examples, which can be identified with epistemic uncertainty, cannot be identified with aleatoric uncertainty alone.

For this we present a unified Bayesian deep learning framework which allows us to learn mappings from input data to aleatoric uncertainty and compose these together with epistemic uncertainty approximations. We derive our framework for both regression and classification applications and present results for per-pixel depth regression and semantic segmentation tasks (see Figure 1 and the supplementary video for examples). We show how modeling aleatoric uncertainty in regression can be used to learn loss attenuation, and develop a complimentary approach for the classification case. This demonstrates the efficacy of our approach on difficult and large scale tasks.

The main contributions of this work are;

1. We capture an accurate understanding of aleatoric and epistemic uncertainties, in particular with a novel approach for classification,

2. We improve model performance by $1 - 3\%$ over non-Bayesian baselines by reducing the effect of noisy data with the implied attenuation obtained from explicitly representing aleatoric uncertainty,

3. We study the trade-offs between modeling aleatoric or epistemic uncertainty by characterizing the properties of each uncertainty and comparing model performance and inference time.

## 2 Related Work

Existing approaches to Bayesian deep learning capture either epistemic uncertainty alone, or aleatoric uncertainty alone [6]. These uncertainties are formalised as probability distributions over either the model parameters, or model outputs, respectively. Epistemic uncertainty is modeled by placing a prior distribution over a model's weights, and then trying to capture how much these weights vary given some data. Aleatoric uncertainty on the other hand is modeled by placing a distribution over the output of the model. For example, in regression our outputs might be modeled as corrupted with Gaussian random noise. In this case we are interested in learning the noise's variance as a function of different inputs (such noise can also be modeled with a constant value for all data points, but this is of less practical interest). These uncertainties, in the context of Bayesian deep learning, are explained in more detail in this section.

### 2.1 Epistemic Uncertainty in Bayesian Deep Learning

To capture epistemic uncertainty in a neural network (NN) we put a prior distribution over its weights, for example a Gaussian prior distribution: $\mathbf{W} \sim \mathcal{N}(0, I)$.

Such a model is referred to as a Bayesian neural network (BNN) [9–11]. Bayesian neural networks replace the deterministic network's weight parameters with distributions over these parameters, and instead of optimising the network weights directly we average over all possible weights (referred to as *marginalisation*). Denoting the random output of the BNN as $\mathbf{f}^{\mathbf{W}}(\mathbf{x})$, we define the model likelihood $p(\mathbf{y}|\mathbf{f}^{\mathbf{W}}(\mathbf{x}))$. Given a dataset $\mathbf{X} = \{\mathbf{x}_1, ..., \mathbf{x}_N\}, \mathbf{Y} = \{\mathbf{y}_1, ..., \mathbf{y}_N\}$, Bayesian inference is used to compute the posterior over the weights $p(\mathbf{W}|\mathbf{X}, \mathbf{Y})$. This posterior captures the set of plausible model parameters, given the data.

For regression tasks we often define our likelihood as a Gaussian with mean given by the model output: $p(\mathbf{y}|\mathbf{f}^{\mathbf{W}}(\mathbf{x})) = \mathcal{N}(\mathbf{f}^{\mathbf{W}}(\mathbf{x}), \sigma^2)$, with an observation noise scalar $\sigma$. For classification, on the other hand, we often squash the model output through a softmax function, and sample from the resulting probability vector: $p(\mathbf{y}|\mathbf{f}^{\mathbf{W}}(\mathbf{x})) = \text{Softmax}(\mathbf{f}^{\mathbf{W}}(\mathbf{x}))$.

BNNs are easy to formulate, but difficult to perform inference in. This is because the marginal probability $p(\mathbf{Y}|\mathbf{X})$, required to evaluate the posterior $p(\mathbf{W}|\mathbf{X}, \mathbf{Y}) = p(\mathbf{Y}|\mathbf{X}, \mathbf{W})p(\mathbf{W})/p(\mathbf{Y}|\mathbf{X})$, cannot be evaluated analytically. Different approximations exist [12–15]. In these approximate inference techniques, the posterior $p(\mathbf{W}|\mathbf{X}, \mathbf{Y})$ is fitted with a simple distribution $q_\theta^*(\mathbf{W})$, parameterised by $\theta$. This replaces the intractable problem of averaging over all weights in the BNN with an optimisation task, where we seek to optimise over the *parameters of the simple distribution* instead of optimising the original neural network's parameters.

Dropout variational inference is a practical approach for approximate inference in large and complex models [15]. This inference is done by training a model with dropout before every weight layer, and by also performing dropout at test time to sample from the approximate posterior (stochastic forward passes, referred to as Monte Carlo dropout). More formally, this approach is equivalent to performing approximate variational inference where we find a simple distribution $q_\theta^*(\mathbf{W})$ in a tractable family which minimises the Kullback-Leibler (KL) divergence to the true model posterior $p(\mathbf{W}|\mathbf{X}, \mathbf{Y})$. Dropout can be interpreted as a variational Bayesian approximation, where the approximating distribution is a mixture of two Gaussians with small variances and the mean of one of the Gaussians is fixed at zero. The minimisation objective is given by [16]:

$$\mathcal{L}(\theta, p) = -\frac{1}{N} \sum_{i=1}^{N} \log p(\mathbf{y}_i|\mathbf{f}^{\widehat{\mathbf{W}}_i}(\mathbf{x}_i)) + \frac{1-p}{2N}||\theta||^2 \tag{1}$$

with $N$ data points, dropout probability $p$, samples $\widehat{\mathbf{W}}_i \sim q_\theta^*(\mathbf{W})$, and $\theta$ the set of the simple distribution's parameters to be optimised (weight matrices in dropout's case). In regression, for example, the negative log likelihood can be further simplified as

$$-\log p(\mathbf{y}_i|\mathbf{f}^{\widehat{\mathbf{W}}_i}(\mathbf{x}_i)) \propto \frac{1}{2\sigma^2}||\mathbf{y}_i - \mathbf{f}^{\widehat{\mathbf{W}}_i}(\mathbf{x}_i)||^2 + \frac{1}{2}\log \sigma^2 \tag{2}$$

for a Gaussian likelihood, with $\sigma$ the model's observation noise parameter – capturing how much noise we have in the outputs.

Epistemic uncertainty in the weights can be reduced by observing more data. This uncertainty induces prediction uncertainty by marginalising over the (approximate) weights posterior distribution.

For classification this can be approximated using Monte Carlo integration as follows:

$$p(y = c | \mathbf{x}, \mathbf{X}, \mathbf{Y}) \approx \frac{1}{T} \sum_{t=1}^{T} \text{Softmax}(\mathbf{f}^{\widehat{\mathbf{W}}_t}(\mathbf{x})) \tag{3}$$

with $T$ sampled masked model weights $\widehat{\mathbf{W}}_t \sim q_\theta^*(\mathbf{W})$, where $q_\theta(\mathbf{W})$ is the Dropout distribution [6]. The uncertainty of this probability vector $\mathbf{p}$ can then be summarised using the entropy of the probability vector: $H(\mathbf{p}) = -\sum_{c=1}^{C} p_c \log p_c$. For regression this epistemic uncertainty is captured by the predictive variance, which can be approximated as:

$$\text{Var}(\mathbf{y}) \approx \sigma^2 + \frac{1}{T} \sum_{t=1}^{T} \mathbf{f}^{\widehat{\mathbf{W}}_t}(\mathbf{x})^T \mathbf{f}^{\widehat{\mathbf{W}}_t}(\mathbf{x}_t) - E(\mathbf{y})^T E(\mathbf{y}) \tag{4}$$

with predictions in this epistemic model done by approximating the predictive mean: $E(\mathbf{y}) \approx \frac{1}{T} \sum_{t=1}^{T} \mathbf{f}^{\widehat{\mathbf{W}}_t}(\mathbf{x})$. The first term in the predictive variance, $\sigma^2$, corresponds to the amount of noise inherent in the data (which will be explained in more detail soon). The second part of the predictive variance measures how much the model is uncertain about its predictions – this term will vanish when we have zero parameter uncertainty (i.e. when all draws $\widehat{\mathbf{W}}_t$ take the same constant value).

### 2.2  Heteroscedastic Aleatoric Uncertainty

In the above we captured model uncertainty – uncertainty over the model parameters – by approximating the distribution $p(\mathbf{W}|\mathbf{X}, \mathbf{Y})$. To capture aleatoric uncertainty in regression, we would have to tune the observation noise parameter $\sigma$.

Homoscedastic regression assumes constant observation noise $\sigma$ for every input point $\mathbf{x}$. Heteroscedastic regression, on the other hand, assumes that observation noise can vary with input $\mathbf{x}$ [17, 18]. Heteroscedastic models are useful in cases where parts of the observation space might have higher noise levels than others. In non-Bayesian neural networks, this observation noise parameter is often fixed as part of the model's weight decay, and ignored. However, when made data-dependent, it can be learned as a function of the data:

$$\mathcal{L}_{\text{NN}}(\theta) = \frac{1}{N} \sum_{i=1}^{N} \frac{1}{2\sigma(\mathbf{x}_i)^2} ||\mathbf{y}_i - \mathbf{f}(\mathbf{x}_i)||^2 + \frac{1}{2} \log \sigma(\mathbf{x}_i)^2 \tag{5}$$

with added weight decay parameterised by $\lambda$ (and similarly for $l_1$ loss). Note that here, unlike the above, variational inference is *not* performed over the weights, but instead we perform MAP inference – finding a single value for the model parameters $\theta$. This approach *does not* capture epistemic model uncertainty, as epistemic uncertainty is a property of the model and not of the data.

In the next section we will combine these two types of uncertainties together in a single model. We will see how heteroscedastic noise can be interpreted as model attenuation, and develop a complimentary approach for the classification case.

## 3  Combining Aleatoric and Epistemic Uncertainty in One Model

In the previous section we described existing Bayesian deep learning techniques. In this section we present novel contributions which extend this existing literature. We develop models that will allow us to study the effects of modeling either aleatoric uncertainty alone, epistemic uncertainty alone, or modeling both uncertainties together in a single model. This is followed by an observation that aleatoric uncertainty in regression tasks can be interpreted as learned loss attenuation – making the loss more robust to noisy data. We follow that by extending the ideas of heteroscedastic regression to classification tasks. This allows us to learn loss attenuation for classification tasks as well.

### 3.1  Combining Heteroscedastic Aleatoric Uncertainty and Epistemic Uncertainty

We wish to capture both epistemic and aleatoric uncertainty in a vision model. For this we turn the heteroscedastic NN in §2.2 into a Bayesian NN by placing a distribution over its weights, with our construction in this section developed specifically for the case of vision models[1].

We need to infer the posterior distribution for a BNN model $\mathbf{f}$ mapping an input image, $\mathbf{x}$, to a unary output, $\hat{\mathbf{y}} \in \mathbb{R}$, and a measure of aleatoric uncertainty given by variance, $\sigma^2$. We approximate the posterior over the BNN with a dropout variational distribution using the tools of §2.1. As before,

we draw model weights from the approximate posterior $\widehat{\mathbf{W}} \sim q(\mathbf{W})$ to obtain a model output, this time composed of both predictive mean as well as predictive variance:

$$[\hat{\mathbf{y}}, \hat{\sigma}^2] = \mathbf{f}^{\widehat{\mathbf{W}}}(\mathbf{x}) \tag{6}$$

where $\mathbf{f}$ is a Bayesian convolutional neural network parametrised by model weights $\widehat{\mathbf{W}}$. We can use a single network to transform the input $\mathbf{x}$, with its head split to predict both $\hat{\mathbf{y}}$ as well as $\hat{\sigma}^2$.

We fix a Gaussian likelihood to model our aleatoric uncertainty. This induces a minimisation objective given labeled output points $x$:

$$\mathcal{L}_{BNN}(\theta) = \frac{1}{D} \sum_i \frac{1}{2} \hat{\sigma}_i^{-2} ||\mathbf{y}_i - \hat{\mathbf{y}}_i||^2 + \frac{1}{2} \log \hat{\sigma}_i^2 \tag{7}$$

where $D$ is the number of output pixels $\mathbf{y}_i$ corresponding to input image $\mathbf{x}$, indexed by i (additionally, the loss includes weight decay which is omitted for brevity). For example, we may set $D = 1$ for image-level regression tasks, or $D$ equal to the number of pixels for dense prediction tasks (predicting a unary corresponding to each input image pixel). $\hat{\sigma}_i^2$ is the BNN output for the predicted variance for pixel $i$.

This loss consists of two components; the residual regression obtained with a stochastic sample through the model – making use of the uncertainty over the parameters – and an uncertainty regularization term. We do not need 'uncertainty labels' to learn uncertainty. Rather, we only need to supervise the learning of the regression task. We learn the variance, $\sigma^2$, implicitly from the loss function. The second regularization term prevents the network from predicting infinite uncertainty (and therefore zero loss) for all data points.

In practice, we train the network to predict the log variance, $s_i := \log \hat{\sigma}_i^2$:

$$\mathcal{L}_{BNN}(\theta) = \frac{1}{D} \sum_i \frac{1}{2} \exp(-s_i) ||\mathbf{y}_i - \hat{\mathbf{y}}_i||^2 + \frac{1}{2} s_i. \tag{8}$$

This is because it is more numerically stable than regressing the variance, $\sigma^2$, as the loss avoids a potential division by zero. The exponential mapping also allows us to regress unconstrained scalar values, where $\exp(-s_i)$ is resolved to the positive domain giving valid values for variance.

To summarize, the predictive uncertainty for pixel $\mathbf{y}$ in this combined model can be approximated using:

$$\text{Var}(\mathbf{y}) \approx \frac{1}{T} \sum_{t=1}^{T} \hat{\mathbf{y}}_t^2 - \left( \frac{1}{T} \sum_{t=1}^{T} \hat{\mathbf{y}}_t \right)^2 + \frac{1}{T} \sum_{t=1}^{T} \hat{\sigma}_t^2 \tag{9}$$

with $\{\hat{\mathbf{y}}_t, \hat{\sigma}_t^2\}_{t=1}^T$ a set of $T$ sampled outputs: $\hat{\mathbf{y}}_t, \hat{\sigma}_t^2 = \mathbf{f}^{\widehat{\mathbf{W}}_t}(\mathbf{x})$ for randomly masked weights $\widehat{\mathbf{W}}_t \sim q(\mathbf{W})$.

### 3.2 Heteroscedastic Uncertainty as Learned Loss Attenuation

We observe that allowing the network to predict uncertainty, allows it effectively to temper the residual loss by $\exp(-s_i)$, which depends on the data. This acts similarly to an intelligent robust regression function. It allows the network to adapt the residual's weighting, and even allows the network to learn to attenuate the effect from erroneous labels. This makes the model more robust to noisy data: inputs for which the model learned to predict high uncertainty will have a smaller effect on the loss.

The model is discouraged from predicting high uncertainty for all points – in effect ignoring the data – through the $\log \sigma^2$ term. Large uncertainty increases the contribution of this term, and in turn penalizes the model: The model *can* learn to ignore the data – but is penalised for that. The model is also discouraged from predicting very low uncertainty for points with high residual error, as low $\sigma^2$ will exaggerate the contribution of the residual and will penalize the model. It is important to stress that this learned attenuation is not an ad-hoc construction, but a consequence of the probabilistic interpretation of the model.

### 3.3 Heteroscedastic Uncertainty in Classification Tasks

This learned loss attenuation property of heteroscedastic NNs in regression is a desirable effect for classification models as well. However, heteroscedastic NNs in classification are peculiar models because technically any classification task has input-dependent uncertainty. Nevertheless, the ideas above can be extended from regression heteroscedastic NNs to classification heteroscedastic NNs.

For this we adapt the standard classification model to marginalise over intermediate heteroscedastic regression uncertainty placed over the *logit space*. We therefore explicitly refer to our proposed model adaptation as a *heteroscedastic* classification NN.

For classification tasks our NN predicts a vector of unaries $\mathbf{f}_i$ for each pixel $i$, which when passed through a softmax operation, forms a probability vector $\mathbf{p}_i$. We change the model by placing a Gaussian distribution over the unaries vector:

$$\hat{\mathbf{x}}_i | \mathbf{W} \sim \mathcal{N}(\mathbf{f}_i^{\mathbf{W}}, (\sigma_i^{\mathbf{W}})^2)$$
$$\hat{\mathbf{p}}_i = \text{Softmax}(\hat{\mathbf{x}}_i). \tag{10}$$

Here $\mathbf{f}_i^{\mathbf{W}}, \sigma_i^{\mathbf{W}}$ are the network outputs with parameters $\mathbf{W}$. This vector $\mathbf{f}_i^{\mathbf{W}}$ is corrupted with Gaussian noise with variance $(\sigma_i^{\mathbf{W}})^2$ (a diagonal matrix with one element for each logit value), and the corrupted vector is then squashed with the softmax function to obtain $\mathbf{p}_i$, the probability vector for pixel $i$.

Our expected log likelihood for this model is given by:

$$\log E_{\mathcal{N}(\hat{\mathbf{x}}_i; \mathbf{f}_i^{\mathbf{W}}, (\sigma_i^{\mathbf{W}})^2)}[\hat{\mathbf{p}}_{i,c}] \tag{11}$$

with $c$ the observed class for input $i$, which gives us our loss function. Ideally, we would want to analytically integrate out this Gaussian distribution, but no analytic solution is known. We therefore approximate the objective through Monte Carlo integration, and sample unaries through the softmax function. We note that this operation is extremely fast because we perform the computation once (passing inputs through the model to get logits). We only need to sample from the logits, which is a fraction of the network's compute, and therefore does not significantly increase the model's test time. We can rewrite the above and obtain the following numerically-stable stochastic loss:

$$\hat{\mathbf{x}}_{i,t} = \mathbf{f}_i^{\mathbf{W}} + \sigma_i^{\mathbf{W}} \epsilon_t, \quad \epsilon_t \sim \mathcal{N}(0, I)$$
$$\mathcal{L}_x = \sum_i \log \frac{1}{T} \sum_t \exp(\hat{x}_{i,t,c} - \log \sum_{c'} \exp \hat{x}_{i,t,c'}) \tag{12}$$

with $x_{i,t,c'}$ the $c'$ element in the logit vector $\mathbf{x}_{i,t}$.

This objective can be interpreted as learning loss attenuation, similarly to the regression case. We next assess the ideas above empirically.

# 4  Experiments

In this section we evaluate our methods with pixel-wise depth regression and semantic segmentation. An analysis of these results is given in the following section. To show the robustness of our learned loss attenuation – a side-effect of modeling uncertainty – we present results on an array of popular datasets, CamVid, Make3D, and NYUv2 Depth, where we set new state-of-the-art benchmarks.

For the following experiments we use the DenseNet architecture [19] which has been adapted for dense prediction tasks by [20]. We use our own independent implementation of the architecture using TensorFlow [21] (which slightly outperforms the original authors' implementation on CamVid by 0.2%, see Table 1a). For all experiments we train with $224 \times 224$ crops of batch size 4, and then fine-tune on full-size images with a batch size of 1. We train with RMS-Prop with a constant learning rate of 0.001 and weight decay $10^{-4}$.

We compare the results of the Bayesian neural network models outlined in §3. We model *epistemic* uncertainty using Monte Carlo dropout (§2.1). The DenseNet architecture places dropout with $p = 0.2$ after each convolutional layer. Following [22], we use 50 Monte Carlo dropout samples. We model *aleatoric* uncertainty with MAP inference using loss functions (8) and (12 in the appendix), for regression and classification respectively (§2.2). However, we derive the loss function using a Laplacian prior, as opposed to the Gaussian prior used for the derivations in §3. This is because it results in a loss function which applies a L1 distance on the residuals. Typically, we find this to outperform L2 loss for regression tasks in vision. We model the benefit of combining both epistemic uncertainty as well as aleatoric uncertainty using our developments presented in §3.

## 4.1  Semantic Segmentation

To demonstrate our method for semantic segmentation, we use two datasets, CamVid [8] and NYU v2 [23]. CamVid is a road scene understanding dataset with 367 training images and 233 test images, of day and dusk scenes, with 11 classes. We resize images to $360 \times 480$ pixels for training and evaluation. In Table 1a we present results for our architecture. Our method sets a new state-of-the-art

| CamVid | IoU |
|---|---|
| SegNet [28] | 46.4 |
| FCN-8 [29] | 57.0 |
| DeepLab-LFOV [24] | 61.6 |
| Bayesian SegNet [22] | 63.1 |
| Dilation8 [30] | 65.3 |
| Dilation8 + FSO [31] | 66.1 |
| DenseNet [20] | 66.9 |
| *This work:* | |
| DenseNet (Our Implementation) | 67.1 |
| + Aleatoric Uncertainty | 67.4 |
| + Epistemic Uncertainty | 67.2 |
| + Aleatoric & Epistemic | **67.5** |

(a) CamVid dataset for road scene segmentation.

| NYUv2 40-class | Accuracy | IoU |
|---|---|---|
| SegNet [28] | 66.1 | 23.6 |
| FCN-8 [29] | 61.8 | 31.6 |
| Bayesian SegNet [22] | 68.0 | 32.4 |
| Eigen and Fergus [32] | 65.6 | 34.1 |
| *This work:* | | |
| DeepLabLargeFOV | 70.1 | 36.5 |
| + Aleatoric Uncertainty | 70.4 | 37.1 |
| + Epistemic Uncertainty | 70.2 | 36.7 |
| + Aleatoric & Epistemic | **70.6** | **37.3** |

(b) NYUv2 40-class dataset for indoor scenes.

Table 1: **Semantic segmentation performance.** Modeling both aleatoric and epistemic uncertainty gives a notable improvement in segmentation accuracy over state of the art baselines.

| Make3D | rel | rms | $\log_{10}$ |
|---|---|---|---|
| Karsch et al. [33] | 0.355 | 9.20 | 0.127 |
| Liu et al. [34] | 0.335 | 9.49 | 0.137 |
| Li et al. [35] | 0.278 | 7.19 | 0.092 |
| Laina et al. [26] | 0.176 | 4.46 | 0.072 |
| *This work:* | | | |
| DenseNet Baseline | 0.167 | 3.92 | 0.064 |
| + Aleatoric Uncertainty | **0.149** | 3.93 | **0.061** |
| + Epistemic Uncertainty | 0.162 | **3.87** | 0.064 |
| + Aleatoric & Epistemic | **0.149** | 4.08 | 0.063 |

(a) Make3D depth dataset [25].

| NYU v2 Depth | rel | rms | $\log_{10}$ | $\delta_1$ | $\delta_2$ | $\delta_3$ |
|---|---|---|---|---|---|---|
| Karsch et al. [33] | 0.374 | 1.12 | 0.134 | - | - | - |
| Ladicky et al. [36] | - | - | - | 54.2% | 82.9% | 91.4% |
| Liu et al. [34] | 0.335 | 1.06 | 0.127 | - | - | - |
| Li et al. [35] | 0.232 | 0.821 | 0.094 | 62.1% | 88.6% | 96.8% |
| Eigen et al. [27] | 0.215 | 0.907 | - | 61.1% | 88.7% | 97.1% |
| Eigen and Fergus [32] | 0.158 | 0.641 | - | 76.9% | 95.0% | 98.8% |
| Laina et al. [26] | 0.127 | 0.573 | 0.055 | 81.1% | 95.3% | 98.8% |
| *This work:* | | | | | | |
| DenseNet Baseline | 0.117 | 0.517 | 0.051 | 80.2% | 95.1% | 98.8% |
| + Aleatoric Uncertainty | 0.112 | 0.508 | 0.046 | 81.6% | 95.8% | 98.8% |
| + Epistemic Uncertainty | 0.114 | 0.512 | 0.049 | 81.1% | 95.4% | 98.8% |
| + Aleatoric & Epistemic | **0.110** | **0.506** | **0.045** | **81.7%** | **95.9%** | **98.9%** |

(b) NYUv2 depth dataset [23].

Table 2: **Monocular depth regression performance.** Comparison to previous approaches on depth regression dataset NYUv2 Depth. Modeling the combination of uncertainties improves accuracy.

on this dataset with mean intersection over union (IoU) score of $67.5\%$. We observe that modeling both aleatoric and epistemic uncertainty improves over the baseline result. The implicit attenuation obtained from the aleatoric loss provides a larger improvement than the epistemic uncertainty model. However, the combination of both uncertainties improves performance even further. This shows that for this application it is more important to model aleatoric uncertainty, suggesting that epistemic uncertainty can be mostly explained away in this large data setting.

Secondly, NYUv2 [23] is a challenging indoor segmentation dataset with 40 different semantic classes. It has 1449 images with resolution $640 \times 480$ from 464 different indoor scenes. Table 1b shows our results. This dataset is much harder than CamVid because there is significantly less structure in indoor scenes compared to street scenes, and because of the increased number of semantic classes. We use DeepLabLargeFOV [24] as our baseline model. We observe a similar result (qualitative results given in Figure 4); we improve baseline performance by giving the model flexibility to estimate uncertainty and attenuate the loss. The effect is more pronounced, perhaps because the dataset is more difficult.

## 4.2 Pixel-wise Depth Regression

We demonstrate the efficacy of our method for regression using two popular monocular depth regression datasets, Make3D [25] and NYUv2 Depth [23]. The Make3D dataset consists of 400 training and 134 testing images, gathered using a 3-D laser scanner. We evaluate our method using the same standard as [26], resizing images to $345 \times 460$ pixels and evaluating on pixels with depth less than $70m$. NYUv2 Depth is taken from the same dataset used for classification above. It contains RGB-D imagery from 464 different indoor scenes. We compare to previous approaches for Make3D in Table 2a and NYUv2 Depth in Table 2b, using standard metrics (for a description of these metrics please see [27]).

These results show that aleatoric uncertainty is able to capture many aspects of this task which are inherently difficult. For example, in the qualitative results in Figure 5 and 6 we observe that aleatoric uncertainty is greater for large depths, reflective surfaces and occlusion boundaries in the image. These are common failure modes of monocular depth algorithms [26]. On the other hand, these qualitative results show that epistemic uncertainty captures difficulties due to lack of data. For

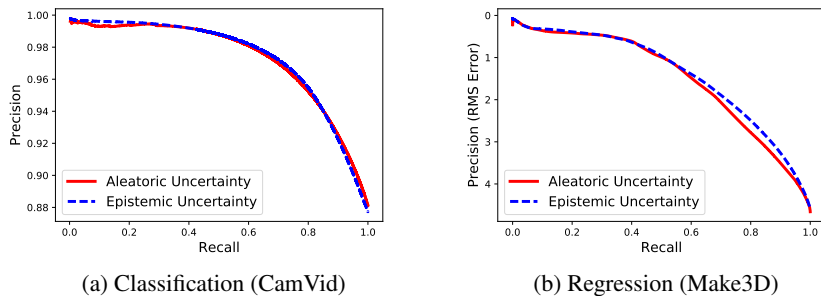

(a) Classification (CamVid)          (b) Regression (Make3D)

Figure 2: Precision Recall plots demonstrating both measures of uncertainty can effectively capture accuracy, as precision decreases with increasing uncertainty.

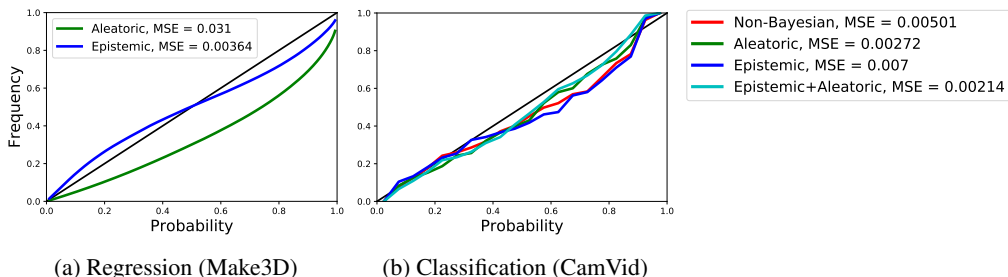

(a) Regression (Make3D)        (b) Classification (CamVid)

Figure 3: Uncertainty calibration plots. This plot shows how well uncertainty is calibrated, where perfect calibration corresponds to the line $y = x$, shown in black. We observe an improvement in calibration mean squared error with aleatoric, epistemic and the combination of uncertainties.

example, we observe larger uncertainty for objects which are rare in the training set such as humans in the third example of Figure 5.

In summary, we have demonstrated that our model can improve performance over non-Bayesian baselines by implicitly learning attenuation of systematic noise and difficult concepts. For example we observe high aleatoric uncertainty for distant objects and on object and occlusion boundaries.

# 5 Analysis: What Do Aleatoric and Epistemic Uncertainties Capture?

In §4 we showed that modeling aleatoric and epistemic uncertainties improves prediction performance, with the combination performing even better. In this section we wish to study the effectiveness of modeling aleatoric and epistemic uncertainty. In particular, we wish to quantify the performance of these uncertainty measurements and analyze what they capture.

## 5.1 Quality of Uncertainty Metric

Firstly, in Figure 2 we show precision-recall curves for regression and classification models. They show how our model performance improves by removing pixels with uncertainty larger than various percentile thresholds. This illustrates two behaviors of aleatoric and epistemic uncertainty measures. Firstly, it shows that the uncertainty measurements are able to correlate well with accuracy, because all curves are strictly decreasing functions. We observe that precision is lower when we have more points that the model is not certain about. Secondly, the curves for epistemic and aleatoric uncertainty models are very similar. This shows that each uncertainty ranks pixel confidence similarly to the other uncertainty, in the absence of the other uncertainty. This suggests that when only one uncertainty is explicitly modeled, it attempts to compensate for the lack of the alternative uncertainty when possible.

Secondly, in Figure 3 we analyze the quality of our uncertainty measurement using calibration plots from our model on the test set. To form calibration plots for classification models, we discretize our model's predicted probabilities into a number of bins, for all classes and all pixels in the test set. We then plot the frequency of correctly predicted labels for each bin of probability values. Better performing uncertainty estimates should correlate more accurately with the line $y = x$ in the calibration plots. For regression models, we can form calibration plots by comparing the frequency of residuals lying within varying thresholds of the predicted distribution. Figure 3 shows the calibration of our classification and regression uncertainties.

| Train dataset | Test dataset | RMS | Aleatoric variance | Epistemic variance |
|---|---|---|---|---|
| Make3D / 4 | Make3D | 5.76 | 0.506 | 7.73 |
| Make3D / 2 | Make3D | 4.62 | 0.521 | 4.38 |
| Make3D | Make3D | 3.87 | 0.485 | 2.78 |
| Make3D / 4 | NYUv2 | - | 0.388 | 15.0 |
| Make3D | NYUv2 | - | 0.461 | 4.87 |

| Train dataset | Test dataset | IoU | Aleatoric entropy | Epistemic logit variance ($\times 10^{-3}$) |
|---|---|---|---|---|
| CamVid / 4 | CamVid | 57.2 | 0.106 | 1.96 |
| CamVid / 2 | CamVid | 62.9 | 0.156 | 1.66 |
| CamVid | CamVid | 67.5 | 0.111 | 1.36 |
| CamVid / 4 | NYUv2 | - | 0.247 | 10.9 |
| CamVid | NYUv2 | - | 0.264 | 11.8 |

(a) Regression        (b) Classification

Table 3: Accuracy and aleatoric and epistemic uncertainties for a range of different train and test dataset combinations. We show aleatoric and epistemic uncertainty as the mean value of all pixels in the test dataset. We compare reduced training set sizes (1, ½, ¼) and unrelated test datasets. This shows that aleatoric uncertainty remains approximately constant, while epistemic uncertainty decreases the closer the test data is to the training distribution, demonstrating that epistemic uncertainty can be explained away with sufficient training data (but not for out-of-distribution data).

## 5.2 Uncertainty with Distance from Training Data

In this section we show two results:

1. Aleatoric uncertainty cannot be explained away with more data,

2. Aleatoric uncertainty does not increase for out-of-data examples (situations different from training set), whereas epistemic uncertainty does.

In Table 3 we give accuracy and uncertainty for models trained on increasing sized subsets of datasets. This shows that epistemic uncertainty decreases as the training dataset gets larger. It also shows that aleatoric uncertainty remains relatively constant and cannot be explained away with more data. Testing the models with a different test set (bottom two lines) shows that epistemic uncertainty increases considerably on those test points which lie far from the training sets.

These results reinforce the case that epistemic uncertainty can be explained away with enough data, but is required to capture situations not encountered in the training set. This is particularly important for safety-critical systems, where epistemic uncertainty is required to detect situations which have never been seen by the model before.

## 5.3 Real-Time Application

Our model based on DenseNet [20] can process a $640 \times 480$ resolution image in $150ms$ on a NVIDIA Titan X GPU. The aleatoric uncertainty models add negligible compute. However, epistemic models require expensive Monte Carlo dropout sampling. For models such as ResNet [4], this is possible to achieve economically because only the last few layers contain dropout. Other models, like DenseNet, require the entire architecture to be sampled. This is difficult to parallelize due to GPU memory constraints, and often results in a $50 \times$ slow-down for 50 Monte Carlo samples.

# 6 Conclusions

We presented a novel Bayesian deep learning framework to learn a mapping to aleatoric uncertainty from the input data, which is composed on top of epistemic uncertainty models. We derived our framework for both regression and classification applications. We showed that it is important to model *aleatoric* uncertainty for:

- **Large data situations**, where epistemic uncertainty is explained away,
- **Real-time applications**, because we can form aleatoric models without expensive Monte Carlo samples.

And *epistemic* uncertainty is important for:

- **Safety-critical applications**, because epistemic uncertainty is required to understand examples which are different from training data,
- **Small datasets** where the training data is sparse.

However aleatoric and epistemic uncertainty models are not mutually exclusive. We showed that the combination is able to achieve new state-of-the-art results on depth regression and semantic segmentation benchmarks.

The first paragraph in this paper posed two recent disasters which could have been averted by real-time Bayesian deep learning tools. Therefore, we leave finding a method for *real-time epistemic uncertainty* in deep learning as an important direction for future research.

## Footnotes

[1]Although this construction can be generalised for any heteroscedastic NN architecture.

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
