[Supplementary Material]

# A  Qualitative Results

Figure 4: NYUv2 40-Class segmentation. From top-left: input image, ground truth, segmentation, aleatoric and epistemic uncertainty.

Figure 5: NYUv2 Depth results. From left: input image, ground truth, depth regression, aleatoric uncertainty, and epistemic uncertainty.

Figure 6: Qualitative results on the Make3D depth regression dataset. Left to right: input image, ground truth, depth prediction, aleatoric uncertainty, epistemic uncertainty.