[Reviews · NeurIPS 2017]

Reviewer 1



This paper proposes a new Bayesian approach to deal with both Aleatoric and Epistemic uncertainties at the same time for deep neural network models. The basic idea is very simple. Basically, the variance of the output is also predicted by the model in addition to the output for aleatoric uncertainty while marginalizing the parameters for epistemic uncertainty. Methods to deal with each uncertainty already exist, but this is the first method to deal with both. The proposed method yields the best results on the two benchmarks they used. The analysis shows the interesting properties of the variances coming from the two uncertainties. This paper is well-written and easy to follow. It has relevant references for previous work. The novelty of the proposed method itself is not very high given that it just additionally predicts the variance of the output for aleatoric uncertainty while using Bayesian Deep Learning for epistemic uncertainty. However, the combination is valuable and this paper gives interesting analysis on the two uncertainties. It is important to understand what uncertainties the current deep learning methods can and cannot deal with. This paper provides insight on that.

Reviewer 2



I have read the other reviews and the rebuttal. Additional comment: The authors asked what I mean by 'numerical uncertainty'. I mean what is called 'algorithmic uncertainty' in this Wikipedia article: https://en.wikipedia.org/wiki/Uncertainty_quantification . Deep learning models are very high dimensional optimisation problems and the optimisers does not converge in almost all cases. Hence, this numerical error, i.e. the difference between the true minimum and the output of the optimiser, is a very significant source of error, which we are uncertain about (because we do not know the solution). Additionally, even rounding errors can add to the uncertainty of deep learning, as is e.g. discussed in this paper: http://proceedings.mlr.press/v37/gupta15.pdf . Hence, I think that the question in the title "What Uncertainties Do We Need in Bayesian Deep Learning for Computer Vision?" is not really answered. Although the paper is apart from this very good, this limits my rating at 'good paper, accept'. In my opinion, the authors should make clearer which kinds of uncertainty they consider (and which they ignore) or why they believe that the uncertainties considered are the dominant ones (which I find completely unclear). % Brief Summary The paper is concerned with evaluating which types of uncertainty are needed for computer vision. The authors restrict their analysis to aleatoric and epistemic uncertainty (leaving out numerical uncertainty). Aleatoric uncertainty includes the uncertainty from statistical noise in data. Epistemic uncertainty is usually another term for ignorance, i.e. things one could in theory know but doesn't know in practice. In this paper however, the authors use epistemic uncertainty as a synonym for model uncertainty (or structural uncertainty, i.e. ignorance over the true physical model which created the data). The paper raises the question how these two sources of uncertainty can be jointly quantified, and when which source of uncertainty is dominant. In particular, this is discussed in the setting of computer vision with deep learning. While classically it was not possible to capture uncertainty in deep learning, Bayesian neural networks (BNN) have made it possible to capture at least epistemic uncertainty. The authors aim to jointly treat epistemic and aleatoric uncertainty in BNNs. To expand the existing literature, the authors propose to include aleatoric uncertainty over pixel output sigma by fixing a Laplace likelihood which contains sigma, which is to be minimised with respect to the parameter vector theta. They also observe that this inclusion of aleatoric uncertainty can be interpreted as loss attenuation. This novel method is then applied to pixel-wise depth regression and semantic segmentation. For semantic segmentation performance, a notable increase in performance is achieved by modelling both aleatoric and epistemic uncertainty. In Chapter 5, an analysis of the effects of epistemic and aleatoric uncertainty follows. Notably, the authors experimentally validate the well-known claims that aleatoric uncertainty cannot be reduced with more data, and that only epistemic uncertainty increases for out-of-data inputs. The paper concludes by claiming that aleatoric uncertainty is particularly important for large data situations and real-time applications, while epistemic uncertainty is particularly important for safety-critical applications and small datasets. % Comment I am skeptical about the use of the classifications of uncertainties which can arise in this setting in this paper. While I agree with the definition of aleatoric uncertainty, I do not agree that epistemic uncertainty is the same as model uncertainty. For me, model uncertainty is the ignorance about the true physical model which creates the data. Epistemic uncertainty is supposed to capture everything which is unknown, but is deterministic and could in theory be known. So, I think that the numerical uncertainty is missing here. In large dimensional optimisation problems (as they arise in deep learning), it is completely unclear whether the optimisation algorithm has converged. I would even expect that the numerical uncertainty could be a dominant source of error, in particular when the computation has too be performed online, e.g. when the self-driving car learns while it's driving. Hence, I would like to ask the authors on their thoughts on this. The second issue I want to arise is the following: The authors convincingly argue that epistemic uncertainty is reduced when more data is available. In statistics, people talk about the statistical consistency in model. A model is statistically consistent when its output is a Dirac measure on the ground truth, in the limit of infinite data. It would be very interesting to see whether statistical consistency results for deep learning (if they exist) can be connected to the paper's pitch on the relative unimportance of epistemic uncertainty in big data situations. Since I think (as explained above) that the classifications of uncertainties is missing the numerical uncertainty of the optimiser, I would argue that the paper does not completely live up to comprehensively answer the question asked in the title. While it is fine, that this is not part in the paper; I think it should be discussed in the introduction. Since, the paper is---a part from this---good, I vote for 'accept'.

Reviewer 3



The paper provides an analysis of the aleatoric and epistemic uncertainty in the Bayesian neural network. It proposes a method that models both types of uncertainty and shows that by doing that the performance of the network could be further improved in real applications such as depth estimation and semantic segmentation. General comment: The problem of uncertainty modeling in representation learning is of great importance, especially to real world safety-critical applications. The paper is well written, technically sound and carefully evaluated. So I recommend accepting the paper. Pros: 1. The paper works on a very important topic of uncertainty modeling in deep learning. 2. The paper is very well written in both text and equations, and easy to understand. 3. The paper introduces a way to combine aleatoric and epistemic uncertainty in Bayesian deep learning, which is shown to outperform state-of-the-art deterministic deep learning models in different real datasets. 4. The paper provides an interesting analysis of aleatoric and epistemic uncertainty based on the observations from the experimental results. Cons: 1. The paper could be further improved by clarifying some of the details: (a) In Equation 6, are the estimations of mean and variance based on the same neural network architecture with some of the parameters shared or independent ones? It seems the prediction and aleatoric uncertainty should be completely isolated. In that case, should they be estimated using completely independent networks? (b) In Table 3(a), the aleatoric variance is getting 20% lower when doing transfer evaluation in NYUv2 while in Table 3(b), the aleatoric variance is about twice of the original value. I wonder if the authors have an explanation of such phenomenon. 2. The paper mentions that the epistemic uncertainty can be explained away by getting a larger training data set. However, the biased (or long-tail) data distribution is an issue in large scale learning. Models learned from long-tail distribution may perform worse in categories with small sample. Also, in that case, the estimation of epistemic uncertainty could be in large variance because the computation of Equation 9 would need a sufficient number of samples. Would the proposed model address such issues?